# Antioxidant and Antibacterial Properties of a Functional Sports Beverage Formulation

**DOI:** 10.3390/ijms24043558

**Published:** 2023-02-10

**Authors:** Alexandros Kolonas, Patroklos Vareltzis, Smaro Kiroglou, Nikolaos Goutzourelas, Dimitrios Stagos, Varvara Trachana, Christina Tsadila, Dimitris Mossialos, Stamatis Mourtakos, Olga Gortzi

**Affiliations:** 1Department of Agriculture Crop Production and Rural Environment, School of Agricultural Sciences, University of Thessaly, 384 46 Volos, Greece; 2Laboratory of Food and Agricultural Industries Technologies, Chemical Engineering Department, Aristotle University of Thessaloniki, 541 24 Thessaloniki, Greece; 3Department of Biochemistry and Biotechnology, School of Health Sciences, University of Thessaly, Biopolis, 415 00 Larissa, Greece; 4Department of Biology, Faculty of Medicine, University of Thessaly, Biopolis, 415 00 Larissa, Greece; 5Microbial Biotechnology-Molecular Bacteriology-Virology Laboratory, Department of Biochemistry & Biotechnology, University of Thessaly, 415 00 Larissa, Greece; 6Department of Nutrition and Dietetics, School of Health Science and Education, Harokopio University of Athens, 176 71 Athens, Greece

**Keywords:** functional foods, beverage development, sports drink, antioxidant activity, antibacterial activity, in vitro digestion

## Abstract

Athletes often consume functional beverages in order to improve performance and reduce oxidative stress caused by high-intensity exercise. The present study aimed to evaluate the antioxidant and antibacterial properties of a functional sports beverage formulation. The beverage’s antioxidant effects were assessed on human mesenchymal stem cells (MSCs) by determining thiobarbituric acid reactive substances (TBARS; TBARS levels decreased significantly by 52.67% at 2.0 mg/mL), total antioxidant capacity (TAC; TAC levels increased significantly by 80.82% at 2.0 mg/mL) and reduced glutathione (GSH; GSH levels increased significantly by 24.13% at 2.0 mg/mL) levels. Furthermore, the beverage underwent simulated digestion following the INFOGEST protocol to assess its oxidative stability. The analysis of the total phenolic content (TPC) using the Folin–Ciocalteu assay revealed that the beverage contained a TPC of 7.58 ± 0.066 mg GAE/mL, while the phenolics identified by HPLC were catechin (2.149 mg/mL), epicatechin (0.024 mg/mL), protocatechuic acid (0.012 mg/mL), luteolin 7-glucoside (0.001 mg/mL), and kaempferol-3-O-β-rutinoside (0.001 mg/mL). The beverage’s TPC was strongly correlated with TAC (R^2^ = 896). Moreover, the beverage showcased inhibitory and bacteriostatic effects against *Staphylococcus aureus* and *Pseudomonas aeruginosa*. Lastly, the sensory acceptance test demonstrated that the functional sports beverage was well accepted by the assessors.

## 1. Introduction

Functional foods are described as foods that can exert physiological effects with health promoting and/or disease preventing properties [1]. The term “functional foods” was first used in the 1980s in Japan, and, despite the fact that there is no internationally acknowledged definition of functional foods, they have become increasingly popular in many countries around the world [1,2]. Functional foods can consist of a single ingredient or a series of ingredients which are not present in similar conventional foods, and they usually include probiotics, prebiotics, phenolic compounds, dietary fibers, vitamins, minerals, phospholipids, proteins, and amino acids [3]. In recent years, functional beverages with health-benefiting ingredients, such as polyphenols and amino acids, have been in high demand for their antioxidant, anti-inflammatory, and anti-aging properties [4]. In addition, functional foods and beverages are also important in sports, as more and more athletes consume them in order to improve exercise performance and reduce oxidative stress [5].

Reactive oxygen species (ROS) are produced continuously in the body via oxidative metabolism, mitochondrial bioenergetics, and immune function, and the most frequent forms of ROS include superoxide anion, hydrogen peroxide, singlet oxygen, hypochlorite, hydroxyl radical, and lipid peroxides, which are involved in the growth, death, and differentiation of cells [6]. Oxidative stress is the result of the imbalance between the production and removal of excess amounts of ROS [6] that can alter cellular compounds, such as lipids, proteins, and DNA, and it is also associated with several diseases, such as cancer, cardiovascular disease, Parkinson’s, and Alzheimer’s [7]. Chronic exercise is known to improve redox balance in humans, and it exerts beneficial effects against the potential risk factors of excessive ROS-mediated diseases [8]. However, high-intensity exercise may lead to oxidative stress, and high levels of ROS appear to cause contractile dysfunction. During exercise, ATP is broken down to release energy, and adenosine monophosphate (AMP) can be formed, which can be further degraded to hypoxanthine, xanthine, and uric acid through a biochemical process involving xanthine oxidase. Xanthine oxidase induces superoxide formation by utilizing molecular oxygen, which results in an increase in ROS production [9]. The intake of antioxidant compounds along with physical exercise may reduce the harmful effects of oxidative stress caused by high-intensity exercise, increase the antioxidant defense associated with exercise, and increase the positive effects of physical activity [7].

There is a growing body of evidence regarding the ergogenic effects of several compounds [5,10,11,12,13,14,15]. L-arginine, a semi-essential amino acid that plays critical roles in muscle development, is widely used as an ergogenic nutrition supplement, and has been investigated for its therapeutic use in a wide variety of pathological conditions, such as hypertension, pulmonary hypertension, angina, nitrate tolerance, and pre-eclampsia [16]. Citrulline is a non-essential amino acid that is metabolized to L-arginine and, when given orally, is more effective than L-arginine in improving arginine plasma concentration because of hepatic first-pass metabolism avoidance and longer circulation time [16]. Citrulline, which is mainly examined in the forms of citrulline malate and L-citrulline, has been studied for its ergogenic, anti-fatigue, and vasodilatory effects [10,14,17,18]. A recent meta-analysis found that citrulline supplementation significantly improved the performance of high-intensity strength and power exercise [14]. Another meta-analysis concluded that citrulline supplementation significantly reduced post-exercise rated perceived exertion (RPE) and muscle soreness without affecting blood lactate levels [18]. Furthermore, it has been shown that citrulline supplementation may increase nitric oxide (NO) production, enhance vasodilation, and improve athletic performance [18]. The most common dosages of citrulline malate and L-citrulline used in clinical trials range from 6 to 12 g and 3 to 6 g, respectively [10,14,18]. In addition, a recent review concluded that a functional beverage that provides the potential ergogenic benefits of L-arginine and L-citrulline supplementation in the convenience of a sports drink is worth considering [17]. Caffeine is the most widely used psychoactive substance and one of the most widely consumed and researched ergogenic aids [11]. Recently, the International Society of Sports Nutrition (ISSN) published a position stand regarding caffeine, in which it is stated that aerobic endurance is the form that benefits most from caffeine use, at doses of 3–6 mg/kg body mass [11]. In addition, a meta-analysis regarding the effect of acute caffeine ingestion on endurance performance found that caffeine can be used effectively to enhance endurance performance, especially in sports where athletes are often separated by small margins [13]. For instance, rowing athletes have been consistently found to benefit from caffeine supplementation [12]. Apart from exercise, caffeine has also been reported to trigger arousal and alertness, improve mood, lead to the release of catecholamines, and even possess antioxidant and anti-inflammatory properties [19].

Hydration and electrolyte requirements constitute another important aspect that needs to be considered in intense exercise. It is widely known that at the end of exhaustive exercise, one should replace the fluids and electrolytes that were lost during exercise. The American College of Sports Medicine (ACSM) also suggests pre-hydrating before the exercise with fluids and electrolytes, such as sodium, potassium, and magnesium [20]. Apart from exercise, magnesium supplementation may reduce blood pressure, hyperglycemia, and hypertriglyceridemia in patients with obesity, type-2 diabetes, and metabolic syndrome [21], while adequate potassium intake plays an important role in achieving lower blood pressure levels [22]. Previously, a single dose containing 3000 mg of sodium di-carbonate, 3000 mg of potassium di-carbonate, 1000 mg of calcium phosphate and calcium citrate, 1000 mg of potassium citrate, and 1000 mg of magnesium citrate significantly improved anaerobic performance in elite soccer players [23].

Polyphenols constitute another important nutrient group that could potentially improve both exercise performance and human health. Polyphenols are phytochemicals with complex structures that have been getting increasing attention due to their biological properties. Specifically, polyphenols are known to possess antioxidant effects, as well as increase the synthesis and bioavailability of NO [24,25,26]. Cocoa (*Theobroma cacao*) powder is a rich source of polyphenols (mainly catechin, epicatechin, anthocyanins, and procyanidins) and has been shown to possess antioxidant and anti-inflammatory properties, increase NO synthesis, induce vasodilation, reduce blood pressure, and regulate lipid synthesis and glucose homeostasis [27]. In addition, a systematic review found that cocoa polyphenol supplementation, containing at least 80 mg of epicatechin per dose, may reduce exercise-induced oxidative stress and improve vascular function [24]. Rosehip (*Rosa canina* L.) is a native shrub that belongs to the *Rosaceae* family, and it is widely consumed in its powder form for its potent antioxidant and anti-inflammatory properties [28,29]. These health benefits are mostly attributed to its polyphenol content, which includes gallic acid, 4-hydroxy benzoic acid, caftaric acid, 2,5-dihidroxy benzoic acid, chlorogenic acid, t-caffeic acid, p-coumaric acid, and ferrulic acid [30]. Strawberries (*Fragaria*) and blueberries (*Vaccinium*) are also rich in polyphenols, such as anthocyanins (which are responsible for the red-blue-purple coloring of berries), flavan-3-ol, ellagitannin, hydroxibenzoic acids, and hydroxycinnamic acids [31,32]. Mainly due to their polyphenol content, the consumption of these fruits has been found to exhibit favorable effects on antioxidant status, inflammation, insulin sensitivity, blood pressure, and endothelial function in human studies [31]. These fruit-derived polyphenols have been studied in concentrations ranging from 50 mg to over 1000 mg and may also enhance exercise performance and recovery via improved muscle perfusion, as well as antioxidant and anti-inflammatory mechanisms [33].

The use of antibiotics—including the overuse and misuse—in human and veterinary practices inevitably have selected for resistant pathogens in the last decades. Therefore, antibiotic resistance is now considered to be one of the top health challenges facing the 21st century that has called the World Health Organization (WHO) into action [34]. Currently, diverse natural products (including functional foods such as honey) demonstrating antibacterial activity have been widely investigated as alternatives in order to prevent infections or to combat multidrug resistant pathogens [35].

Even though sports and energy drinks are widely consumed, their safety is controversial and their over-consumption has been linked with several side-effects. Specifically, it has been shown that they can cause insomnia, nervousness, stress and dehydration [36,37,38]. In addition, they possess dental erosive potential which can lead to poor dental health, mainly due to the high sugar content [38,39]. Lastly, frequent consumption of sports and energy drinks can increase the risk for chronic diseases, such as obesity and type 2 diabetes [36,38]. In recent years, there have been several attempts to develop functional drinks with health benefits but, to the best of our knowledge, none of them aimed at simultaneously improving exercise performance [40,41,42,43]. Therefore, the objectives of this study were (a) to develop a formulation of a functional sports beverage with ergogenic and antioxidant properties, (b) to investigate the effect of simulated gastrointestinal digestion on the oxidative stability of the beverage formulated, (c) to analyze its antioxidant and antibacterial activity, and (d) to evaluate the sensory acceptability of the beverage formulated.

## 2. Results and Discussion

### 2.1. Total Phenolic Content

The total phenolic content (TPC) of the functional sports beverage was found to be 7.58 ± 0.066 mg GAE/mL. This relatively high phenolic content of the powder may be due to the phenolic compounds found in the cocoa extract, the mixed berries powder, and the rosehip powder. Previous studies have shown that cocoa has a total flavanol content that ranges from 1.32 ± 0.17 to 2.73 ± 0.18 mg GAE/g dry matter (DM), with epicatechin being the main flavanol (0.57 ± 0.07 to 1.15 ± 0.06 mg GAE/g DM) [44]. Furthermore, a functional beverage containing cocoa bean shell was found to have a TPC content that ranged from 0.13 mg GAE/mL to 1.8 mg GAE/mL [45]. In addition, blueberry juice has been found to possess a TPC ranging from 3.56 ± 0.15 to 8.52 ± 0.19 mg GAE/mL [46]. Similarly, strawberry juice has been shown to have a TPC ranging from 6.05 ± 0.17 to 13.8 ± 0.3 mg GAE/mL [47]. Lastly, the TPC of rosehip has been shown to vary from 10.89 ± 0.54 mg GAE/g DM to 26.49 ± 1.32 mg GAE/g DM [48].

### 2.2. Polyphenolic Composition by HPLC

From the HPLC analysis, five main polyphenolic compounds were identified. Peak 1 was identified as epicatechin, peak 2 as protocatechuic acid, peak 3 as catechin, peak 4 as luteolin-7-glucoside, and peak 5 as kaempferol-3-O-β-rutinoside (Figure 1). As can be seen in Table 1, the amounts of the identified compounds were 23.01 mg/g or 2.149 mg/mL for catechin, 0.26 mg/g or 0.024 mg/mL for epicatechin, 0.13 mg/g or 0.012 mg/mL for protocatechuic acid, 0.01 mg/g or 0.001 mg/mL for luteolin 7-glucoside, and 0.01 mg/g or 0.001 mg/mL for kaempferol-3-O-β-rutinoside regarding the powder (before dissolving in water) and the beverage (after dissolving the powder in water), respectively. The limit of detection (LOD) and limit of quantification (LOQ) were established experimentally using the ratio signal-to-noise methodology. Due to the fact that all considered compounds, except for kaempferol-3-O-β-rutinoside, appeared in the chromatogram with high peaks, the LOD and LOQ were determined only for kaempferol-3-O-β-rutinoside (0.01 mg/g), and the results obtained for these limits were 0.002 mg/g and 0.006 mg/g, respectively.

The main compounds identified in this work are in line with previous findings [49,50,51,52,53]. Specifically, blueberries, strawberries and blackberries have been shown to contain catechin, luteolin, protocatechuic acid, and kaempferol [49,50,51]. In addition, the main phenolic compounds identified in cocoa are epicatechin and catechin [52]. Lastly, HPLC analyses have identified catechin and kaempferol in rosehip [53].

### 2.3. In Vitro Antioxidant Activity

#### 2.3.1. Determination of Extracts’ Non-Cytotoxic Concentrations in MSC Cells

Before examining the beverage’s antioxidant effects in MSC cells, its cytotoxicity was determined using the XTT assay in order to select non-cytotoxic concentrations. The results showed that the beverage had no significant cytotoxicity at concentrations up to 2 mg/mL (Figure 2).

#### 2.3.2. Effects of Powder on TAC, TBARS and GSH Levels in MSC Cells

The results showed that the beverage treatment significantly increased TAC levels by 62.83 and 80.82% at 1.0 and 2.0 mg/mL, respectively, in MSC cells (Figure 3). Moreover, TBARS levels in MSC cells treated with the beverage were decreased significantly by 58.65 and 52.67% at 1.0 and 2.0 mg/mL compared to the control, respectively (Figure 3). Finally, the beverage treatment significantly increased GSH levels by 27.41 and 24.13% at 1.0 and 2.0 mg/mL in MSC cells, respectively (Figure 3).

Flavonoid-rich dark chocolate consumption has been shown to significantly decrease TBARS, while procyanidin-rich chocolate led to a decrease of 40% in plasma TBARS and 31% in plasma TAC [54]. In addition, cocoa treatment has been found to significantly increase GSH levels in liver tissue and heart tissue of male Albino Wistar rats [55]. Furthermore, chocolate containing raspberry and blueberry extracts led to significantly decreased TBARS levels [56]. Moreover, our research group has reported that methanolic extracts of Rosa canina decreased ROS and increased GSH levels in human endothelial cells [57]. Lastly, it has been found that plant-derived exosome-like nanovesicles from strawberries can prevent oxidative stress in MSC cells, possibly due to their phenol and vitamin C content [58].

When examining the bioactivities of food or beverages, the absorption and the bioavailability of their bioactive compounds are an important issue. Several bioactive compounds have been found to be effective in in vitro studies, but at much higher concentrations than those of their peak levels in human plasma [59,60]. In the present study, the maximum concentration of the beverage in the form of powder used in the cell culture experiments was 2 mg/mL. The main polyphenolic compounds identified in the beverage were catechin, epicatechin, protocatechuic acid, luteolin-7-glucoside, and kaempferol-3-O-rutinoside. According to the amount of these polyphenols in the beverage, their concentrations in the cell culture medium at 2 mg/mL of the beverage in the form of powder were catechin (150 μΜ), epicatechin (1.8 μΜ), protocatechuic acid (1.7 μΜ), luteolin-7-glucoside (0.047 μΜ), and kaempferol-3-O-rutinoside (0.034 μΜ). The concentration of epicatechin was physiologically meaningful, since it has been shown to achieve 0.6 μΜ in human plasma after the administration of green tea [61]. Protocatechuic acid’s concentration can also be achieved in the human organism, since this phenolic acid has been reported to remain stable in human plasma at 30 μΜ for 24 h [62]. Moreover, the concentration of luteolin-7-glucoside was also meaningful, since a bioavailability study has reported that it absorbed quickly and reached levels of about 0.7 μΜ in rat plasma [63]. Regarding kaempferol-3-O-rutinoside’s bioavailability, studies have shown that kaempferol glucosides are mainly absorbed through mechanisms of active transport and deglycosylation [64]. However, there has not been much data on the bioavailability of kaempferol and its glycosides forms. A concentration of about 0.05 mM of this polyphenol has been found in human plasma [64]. Thus, our concentration of kaempferol-3-O-rutinoside of 0.034 μΜ may be found in the human organism. Catechin’s concentration of 150 μΜ in our beverage formulation was higher than that reported in human plasma after the consumption of beverages that were rich in this polyphenol [60].

#### 2.3.3. Total Antioxidant Capacity (TAC) and Its Correlation with the Total Phenolic Content

The correlation between TAC and the total phenolic content is shown in Figure 4. The correlation coefficient was R^2^ = 0.896 (y = 4.3246x + 119.21). This result suggests that the antioxidant activity has a positive correlation with the total phenolic content and indicates that approximately 89% of the antioxidant activity of the beverage possibly resulted from its content in phenolic compounds. However, it can also be suggested that the antioxidant activity of the beverage possibly derived from the rest of the compounds it contained, such as caffeine anhydrous or citrulline malate, and was not limited to phenols. Previous studies have shown that the antioxidant activity of phenols is possibly due to their capabilities to act as reducing agents and hydrogen donors [65].

For examining the beverage’s antioxidant activity, its ability to either enhance antioxidant mechanisms or decrease ROS-induced damage was assessed in human MSCs. The results showed that the beverage treatment increased the TAC in MSCs, thus indicating the enhancement of antioxidant molecules. Specifically, a TAC assay based on DPPH^●^ measured the antioxidant activity of non-enzymatic molecules acting either as hydrogen atom transfers (HAT) or single electron transfers (SET) [66]. One of the most important non-enzymatic antioxidant molecules within cells is GSH [67]. Thus, the beverage-induced increase in GSH levels in MSCs was in accordance with the observed increase in the TAC. Beverage treatment may cause GSH increases by activating enzymes involved in GSH synthesis (e.g., glutamate cysteine synthase and glutathione synthetase). Otherwise, the beverage may increase GSH levels through protection from oxidation by free radical scavenging. As shown in the DPPH assay, the beverage by itself possessed compounds having free radical scavenging activity. In a different manner, the beverage could augment other antioxidant molecules scavenging free radicals, as suggested by the TAC increase. Moreover, beverage treatment decreased TBARS levels, a marker of lipid peroxidation, in MSCs. This TBARS’ decrease could be attributed, at least in part, to beverage-induced increases in TAC and GSH. The beverage-induced decrease in TBARS was important, since TBARS is a marker of lipid peroxidation, which is a self-propagating chain reaction. When lipid peroxidation occurs, damage may be caused to different cellular parts, especially extra- and intracellular membranes, which can even lead to cellular death [68].

Cell death of stem cells such as MSCs is a decisive factor for pathologies such as aging [69]. Moreover, in aging, as cell damage accumulates over time, increased ROS’ levels maintain cell survival [69]. However, this adaptive mechanism has a limit, and so when ROS’ homeostatic role is surpassed, this deteriorates cellular damage and, subsequently, the aging process [69]. Moreover, MSCs’ use has been proposed for the treatment of several diseases such as cardiovascular and neurodegenerative, and, as a result, they have a central role in regenerative medicine [70,71,72]. However, oxidative stress leads to MSCs’ death, and thus impairs their therapeutic effects [73]. Therefore, identifying antioxidant compounds, such as the formulated beverage, which can improve stem cells’ redox status, is of great importance [74].

### 2.4. Antibacterial Activity

The antibacterial activity of the beverage against methicillin-resistant *Staphylococcus aureus* 1552 and carbapenem-resistant *Pseudomonas aeruginosa* 1773 is shown in Table 2. The beverage showed inhibitory effects against both tested bacterial strains at a concentration of 30 mg/mL. However, the beverage exerted only bacteriostatic, not bactericidal effects against these strains at tested concentrations (up to 30 mg/mL). This is in accordance with previous findings regarding some of the ingredients used in this formulation, which demonstrated that strawberries’ aqueous extract exerted bacteriostatic effects against *Staphylococcus aureus* and *Pseudomonas aeruginosa* (both at 10% *w*/*v*) but not bactericidal activity [75]. The same study reported that blueberries’ aqueous extract and raspberries’ aqueous extract exhibited inhibitory effects against *Staphylococcus aureus* (5% *w*/*v*) and *Pseudomonas aeruginosa* (5% *w*/*v*), but only the aqueous extract of blueberries showed bactericidal effects against both strains at a concentration of 10% *w*/*v* [75]. Furthermore, the MICs of various cocoa powder extracts against *Staphylococcus aureus* and *Pseudomonas aeruginosa* have been reported to range from 5 to 25 mg/mL and from 7.5 to 25 mg/mL, respectively [76]. The methanolic extract of cocoa pod husk has shown inhibitory and bactericidal activity against *Staphylococcus aureus* at a concentration of 0.62 mg/mL and 5 mg/mL, respectively [65]. In addition, pure caffeine has been shown to exert inhibitory effects against *Staphylococcus aureus* at a concentration of 200 μg/mL [77]. These differences might be attributed to the relatively low concentrations of both cocoa extract (0.2% *w*/*v*) and caffeine anhydrous (0.05% *w*/*v*) in the beverage formulation.

### 2.5. Oxidative Stability

The beverage was exposed to simulated gastrointestinal conditions (INFOGEST protocol), and the oxidative stability was followed at each stage (oral digestion phase, gastric digestion phase, intestinal digestion phase) by % DPPH scavenging and ferric reducing antioxidant power (FRAP). The oxidative stability of the beverage is shown in Table 3. The results suggest that the simulated digestion conditions did not negatively affect the oxidative stability and the antioxidant potential of the beverage. On the contrary, it was found that the % DPPH scavenging of the beverage significantly increased from the oral digestion phase (70.8 ± 1.7% DPPH scavenging) to the gastric digestion phase (74.8 ± 1.3% DPPH scavenging) to the intestinal digestion phase (88.1 ± 0.3% DPPH scavenging). Similarly, the FRAP significantly increased from the oral digestion phase and gastric digestion phase (728 ± 93 and 804 ± 180 μΜ, respectively) to the intestinal digestion phase (1855 ± 72 μM). Other studies have also shown an increase in antioxidant potential of foods during gastrointestinal digestion. This has been attributed to the release of extra phenols from the food matrix, especially in the intestinal phase, due to the action of proteases and lipases [78,79]. It is possible that phenolics bound to the insoluble fiber of berries and rosehip powders were released during digestion and, therefore, increased the antioxidant potential of the beverage [80].

Similarly to our findings, previous studies have shown that the ferric reducing antioxidant power of rosehip varies from 526.67 ± 26.33 μM of trolox equivalents (TE/g) to 1774.51 ± 88.83 μM TE/g [48,49]. In addition, rosehip extract at concentrations of 2.5, 5.0, and 10.0 mg/mL scavenged DPPH by 39.43, 59.71, and 96.54%, respectively [81]. Cocoa powder extracts have been shown to have an antioxidant capacity from 11.33 to 26.33 μM/L Fe II in a FRAP assay and from 9.47 to 11.68 μM/L Trolox in a DPPH assay [82]. In addition, cocoa polyphenol extract at a concentration of 1 mg/mL has been shown to scavenge DPPH by 78.34% [83]. Lastly, wild blueberries have been shown to possess FRAP values of 43.97 ± 8.93 μΜ/g, while wild blackberries have a FRAP value of 59.36 μΜ/g [84].

### 2.6. Sensory Acceptance

The sensory profile of the beverage is illustrated in the spider web chart in Figure 5. Attributes with a mean score of >6 are highly preferred or liked, while attributes with a mean score <4 are unacceptable. The attribute with the lowest mean score was solubility (7.13 ± 0.88), and the attribute with the highest mean score was aroma (8.23 ± 0.67), while the mean score for the overall acceptance was 7.42 ± 0.89. These results suggest that the beverage was well-accepted regarding all nine attributes (taste, aftertaste, color, aroma, solubility, mouthfeel, acidity, sweetness, and overall acceptance) among the assessors.

The results of the short questionnaire that the assessors filled out regarding their demographic characteristics, the frequency at which they consume sports/energy drinks, and the frequency at which they engage in physical activity/exercise are shown in Table 4. The sensory acceptance test included 31 (16 male, 15 female) assessors, of whom 21 (67.7%) where in the 18–29 age group, 7 (22.6%) in the 30–44 age group, and 3 (9.7%) where in the 45+ age group. The questionnaire results showed that 96.8% (N = 30) of the assessors would be willing to buy the beverage that they tested and that they would prefer it over others on the market. Lastly, 74.2% (N = 23) of the assessors described the beverage’s flavor as “berries”, while 25.8% (N = 8) of the assessors described the beverage’s flavor as “strawberry”.

Sports and energy drinks are widely consumed, but their consumption has been linked with several side-effects, such as insomnia, nervousness, stress, dehydration and poor dental health [36,37,38]. In addition, both sports and energy drinks contain high amounts of sugars, while there is convincing evidence linking sugar-sweetened beverages to an increased risk for type 2 diabetes and obesity [38]. Specifically, many sports drinks contain 15 g of sugar per serving (237 mL), and many energy drinks contain 35 g of sugar per serving (237 mL) [38].

In recent years, functional beverages with health-promoting properties have been the center of attention for their antioxidant, anti-inflammatory, and anti-aging properties [4]. These beverages have been getting increasing attention from athletes, who consume them in order to improve exercise performance and reduce oxidative stress [6,10], since high-intensity exercise can lead to temporarily increased oxidative stress and cause contractile dysfunction [7]. The intake of antioxidant compounds can reduce oxidative stress, and it has even been suggested that a functional beverage formulation, which provides the potential ergogenic benefits of citrulline supplementation in the form of a sports drink, is worth considering [17].

In this study, a functional sports beverage, without added sugars, in the form of a powder that can be dissolved in water, was developed. The final formulation contained citrulline malate (2.4% *w*/*v*), magnesium citrate (0.4% *w*/*v*), tripotassium citrate (0.4% *w*/*v*), trisodium citrate (0.4% *w*/*v*), mixed berries powder (0.3% *w*/*v*), rosehip powder (0.2% *w*/*v*), cocoa extract (0.2% *w*/*v*), caffeine anhydrous (0.05% *w*/*v*), steviol glycosides (3% *w*/*v*), and maltodextrin (0.2% *w*/*v*). Citrulline has been extensively studied at doses ranging from 3 to 12 g and has repeatedly exhibited endurance-enhancing, power-enhancing, vasodilatory, and anti-fatigue effects [14,17,18]. Taking this into account, the formulated beverage contains citrulline malate at a concentration of 2.4% *w*/*v* or 6 g per serving, which should be optimal in order to exhibit its ergogenic effects. In addition, caffeine, at concentrations of 3–6 mg/kg body mass, has been repeatedly shown to be one of the most effective and widely-used ergogenic aids in order to improve exercise performance [11,13]. The formulated beverage contains caffeine anhydrous at a concentration of 0.05% or 125 mg per serving, which should suffice in order to exhibit its ergogenic effects without the adverse effects, such as heart rate variability and sleep latency [85]. In addition, the formulated beverage contains tripotassium citrate, trisodium citrate, and magnesium citrate in order to achieve optimal pre-hydration, according to the recommendations of the ACSM [20].

The present study showcased the relatively high total phenol content of the formulated beverage, with the main phenolic compounds identified being catechin, epicatechin, protocatechuic acid, luteolin 7-glucoside, and kaempferol-3-O-β-rutinoside, as well as high oxidative stability through the simulated gastrointestinal conditions. In addition, the beverage’s antioxidant properties positively correlated with the total phenol content (R^2^ = 0.896), which suggests that approximately 89% of its antioxidant properties can be attributed to its content in phenolic compounds. Furthermore, in order to sustain a low sugar content, stevia was used as a sweetener, which has been shown to enhance the bioavailability of polyphenols [86]. In addition, cocoa extract was added to the beverage, which has been shown to be able to reduce exercise-induced oxidative stress, due to the improvement of the nitric oxide function [24,87]. Lastly, the beverage contains rosehip powder and mixed berries powder. Fruit-derived polyphenols have been shown to improve exercise performance and enhance recovery by improving vascular function via NO-mediated mechanisms, and by decreasing oxidative stress via signaling through the nuclear factor erythroid 2–related factor 2 (Nrf2) [33].

Thus, the composition of the formulated beverage could potentially exhibit ergogenic and anti-fatigue properties, while the oxidative properties of the beverage could potentially contribute to mediating exercise-induced oxidative stress. Further experiments and clinical trials need to be conducted in order to evaluate the ergogenic effects of this beverage in athletes.

## 3. Materials and Methods

### 3.1. Development of the Beverage

#### 3.1.1. Materials

Food-grade citrulline malate 2:1 (THG plc., Manchester, UK), magnesium citrate (THG plc., Manchester, UK) tripotassium citrate (Manis Chemicals, Athens, Greece), trisodium citrate (Manis Chemicals, Athens, Greece), cocoa extract (Foodspring GmbH, Berlin, Germany), rosehip powder (Health Trade O.E., Achaia, Greece), mixed frozen berries (blueberries 20%, strawberries 20%, raspberries 20%, black gooseberries 20%, red gooseberries 20%; Lidl Hellas & SIA OE., Athens, Greece), maltodextrin (Manis Chemicals, Athens, Greece), caffeine anhydrous (Manis Chemicals, Athens, Greece), and steviol glycosides (Only Bio, Athens, Greece) were bought and used to prepare beverage samples.

#### 3.1.2. Preparation of Mixed Berries Powder

The mixed frozen berries were used to create a berry powder that would play a three-fold role in the final beverage: (a) enhance the antioxidant activity of the beverage, (b) serve as the colorant of the beverage, and (c) serve as the flavoring of the beverage. In order to prepare the mixed berries powder, 750 g of mixed frozen berries were added to a blender (Munro 800 S, Essex, UK) and blended until homogenized. Then, the blended berries were dehydrated in a HENDI 229033 dehydrator (PKS HENDI South East Europe SA, Athens, Greece) for 48 h at 50 °C. Afterwards, the dehydrated berries were ground using a Bosch TSM6A013B grinder (Robert Bosch GmbH, Stuttgart, Germany), and the berry powder was sieved in order to remove coarser particles. Finally, the berry powder was stored in an airtight container. From 750 g of mixed frozen berries, we retrieved 75 g of mixed berries powder.

#### 3.1.3. Beverage Formulation

Several beverage formulations were initially developed in order to reach the effective concentration of the compounds, while sustaining the solubility and the desired organoleptic characteristics of the beverage, using laboratory scale trials involving staff and students in the food lab as panelists to evaluate the formulations. The final beverage formulation contains citrulline malate (2.4% *w*/*v*), magnesium citrate (0.4% *w*/*v*), tripotassium citrate (0.4% *w*/*v*), trisodium citrate (0.4% *w*/*v*), mixed berries powder (0.3% *w*/*v*), rosehip powder (0.2% *w*/*v*), cocoa extract (0.2% *w*/*v*), caffeine anhydrous (0.05% *w*/*v*), steviol glycosides (3% *w*/*v*), and maltodextrin (0.2% *w*/*v*). This formulation was in the form of an instant beverage; viz., in the form of powder that is dissolved in water. Specifically, one serving of the final beverage is equal to 23.35 g of the powder dissolved in 250 mL of water.

### 3.2. Determination of Total Phenolic Content (Folin-Ciocalteu)

Polyphenols were photometrically determined by the Folin–Ciocalteu procedure according to Gortzi et al. [88] with slight modifications. A calibration curve was prepared using the absorbance of 5 standard solutions (with concentrations of 1.0–10 mg/L of gallic acid) prepared in 25-mL flasks. During the preparation of these solutions, 1 mL of Folin–Ciocalteu reagent (AppliChem, ITW Reagents) was added in each flask. After 3 min, 2 mL of Na_2_CO_3_ aqueous solution (25% *w*/*v*) was added. The flasks were then filled with DI water and left to stand for 1 hr in the dark. Finally, the absorbance of each solution was measured at 725 nm using a JASCO V-630 spectrophotometer. A solution including all of the reagents without the addition of gallic acid was used as a blank. For the polyphenol determination of the sample, 0.934 g of the beverage powder were mixed with 5 mL MeOH and 5 mL DI water, and 1 mL of this solution was added in a 25 mL flask containing 15 mL DI water. Then, 1 mL of Folin–Ciocalteu reagent was added and, after 3 min, 2 mL of Na_2_CO_3_ aqueous solution (25% *w*/*v*) were added, and the flask was filled with DI water. Finally, their concentration was photometrically determined using the calibration curve (y = 0.0921x + 0.0496 and R^2^ = 0.983). The concentration of the total phenolic content in the beverage was reported as milligrams of gallic acid equivalents (GAE) per mL.

### 3.3. HPLC

The method was conducted according to the method by Athanasiadis et al. [89]. Briefly, a Shimadzu liquid chromatograph (CBM-20A) and a Shimadzu detector (SPD-M20A) were used. A Phenomenex Luna C18(2) (100 Å, 5 μm, 4.6 × 250 mm) (Phenomenex, Inc., Torrance, CA, USA) retained at 40 °C, a flow rate of 1 mL min^−1^, and an injection volume of 20 μL were used. The mobile phases and the elution program used have been described previously [90]. Quantification calibration curves were prepared using three points (0, 10, and 50 mg mL^−1^), for caffeic acid (quantified at 320 nm, y = 0.000009x + 0.8755, R^2^ = 0.9986), rosmarinic acid (at 320 nm, y = 0.00002x + 0.3334, R^2^ = 0.9998), and luteolin-7-O-glucoside (at 345 nm, y = 0.00002x + 1.0794, R^2^ = 0.9980). The estimation of the total area was carried out at 280 nm and 345 nm.

### 3.4. In Vitro Antioxidant Activity

#### 3.4.1. Cell Culture Conditions

Human mesenchymal stem cells (MSCs) were obtained from the Wharton jelly of umbilical cords (WJ-MSCs) from term gestation newborns after birth, having obtained consent from the parents, as previously described [91]. Isolated WJ-MSCs were cultured, as reported previously [69], in Dulbecco’s modified Eagle’s medium (DMEM) high glucose with stable glutamine and sodium pyruvate (BioWest, Miami, FL, USA) plus 10% fetal bovine serum (FBS; Thermo Fisher Scientific, Waltham, MA, USA) and 1% penicillin/streptomycin (Thermo Fisher Scientific, Waltham, MA, USA) at 37 °C in a humidified atmosphere of 5% CO_2_. Cells were maintained in culture in order to produce three different passages (i.e., p18, p23, and p28) that were used for the experiments. The medium was changed twice a week and cells were passed when 90% confluency was reached.

#### 3.4.2. XTT Cell Viability Assay

The antioxidant activity of the beverage in MSC cells was examined using non-cytotoxic concentrations. In order to select these concentrations, the cytotoxicity of the beverage was determined using the XTT cell viability assay kit (Sigma) as previously described [92]. Briefly, MSC cells were seeded into a 96-well plate (1 × 10^4^ cells per well) in DMEM containing 10% fetal bovine serum (FBS). After 24 h incubation at 37 °C in 5% CO_2_, cells were treated with different concentrations of the beverage in DMEM plus 10% FBS and incubated for another 24 h. Then, 50 µL of XTT test solution were added to each well. After 4 h of incubation, absorbance was measured at 450 nm and also at 630 nm as a reference wavelength in a Bio-Tek ELx800 microplate reader (Winooski, VT, USA). The negative control was cells in DMEM plus 10% FBS. Also, the absorbance of each beverage concentration alone in DMEM plus 10% FBS and XTT test solution was tested at 450 nm. The absorbance values shown by the beverage alone were subtracted from those derived from cell treatment with the beverage. The absorbance values of the control and the beverage were used for calculating the percentage inhibition of cell growth caused by the beverage treatment using the following formula:Inhibition (%) = [(O.D.control − O.D.sample)/O.D.control] × 100
where O.D.control and O.D.sample indicate the optical density of the negative control and the tested substance, respectively. All experiments were carried out in triplicate and on three separate occasions.

#### 3.4.3. Cell Treatment with Beverage

For assessing the beverage’s effect on the MSCs’ redox status, cells were seeded into 75 cm^2^ flasks in DMEM containing 10% FBS. After 24 h incubation at 37 °C in 5% CO_2_, the cells were treated with different concentrations of the beverage formulation in the form of powder in DMEM, with 10% FBS, and again incubated for 24 h. Then, cells were detached by trypsinization and used for the determination of lipid peroxidation, total antioxidant capacity (TAC), and reduced glutathione (GSH) levels.

#### 3.4.4. TBARS Assay for the Determination of Lipid Peroxidation

For the determination of lipid peroxidation, TBARS assay was used. After trypsinization, the cells were suspended in PBS buffer and lysed by vortexing. Afterwards, cell lysates’ protein concentration was determined using Bradford assay. Then, a slightly modified TBARS assay, as described by Keles et al. [93], was used. Specifically, a total of (400-X) µL of PBS, where X was the amount of cell suspension to have 30 µg of protein in the assay, or 400 µL of PBS for the blank, was mixed with 500 µL Tris-HCl (200 mM, pH 7.4) and 500 µL 35% TCA and incubated for 10 min at room temperature. Afterwards, a 1 mL solution consisting of 2 M Na_2_SO_4_ and 55 mM thiobarbituric acid was added, and the samples were incubated at 95 °C for 45 min. Subsequently, the samples were cooled on ice for 5 min and vortexed following the addition of 1 mL of 70% TCA. The samples were then centrifuged at 15,000× *g* for 3 min, and the absorbance of the supernatant was read at 530 nm. The calculation of the TBARS concentration was based on the molar extinction co-efficient of malondialdehyde and was expressed as TBARS nmol per mg of protein of cell lysate. Each experiment was repeated at least three times.

#### 3.4.5. Assessment of TAC Levels

After trypsinization, the cells were suspended in PBS buffer and lysed using sonication, and cell lysates’ protein concentrations were determined using Bradford assay. Then, TAC was assessed as described previously [94]. The reaction was performed in 1 mL containing 50 μL of cell lysate containing 30 μg of protein, 450 μL of 10 mM sodium phosphate buffer (pH = 7.4), and 500 μL of 0.1 mM 2,2-diphenyl-1-picrylhydrazyl (DPPH⋅) radical solution. Samples containing only the radical solution were diluted in sodium phosphate buffer (pH = 7.4) and were used as control. The samples were mixed vigorously and incubated for 60 min at room temperature (RT) in the dark. Then, they were centrifuged (20,000× *g*, 3 min, 4 °C), and the absorbance was monitored at 517 nm. TAC was expressed as μmol DPPH⋅ reduced to 1,1-diphenyl-2-picryldrazine (DPPH-H) by the antioxidant components of the cell lysate per mg of sample protein. Each experiment was repeated at least three times.

#### 3.4.6. Assessment of GSH Levels

After trypsinization, the cells were suspended in PBS buffer and lysed using sonication. Afterwards, cell lysates’ protein concentrations were determined using Bradford assay. Then, GSH levels were assessed as described previously [94]. In particular, the reaction was performed in 1 mL containing 520 μL of 67 mM sodium phosphate buffer (pH = 8.0), 150 μL of cell lysate suspension containing 30 μg protein, and 330 μL of 1 mM 5,5′-dithiobis (2-nitrobenzoic acid) (DTNB) solution. The samples were mixed and incubated at RT in the dark for 15 min, and the absorbance was monitored at 412 nm. The concentration of GSH was calculated on the basis of the millimolar extinction coefficient of DTNB and was expressed as nmol GSH/mg of protein of cell lysate. Each experiment was repeated at least three times.

### 3.5. Antibacterial Activity

#### 3.5.1. Bacterial Strains and Growth Conditions

The antibacterial activity was tested against methicillin-resistant *Staphylococcus aureus* 1552 and carbapenem-resistant *Pseudomonas aeruginosa* 1773. These clinical strains were identified and characterized by standard methods (kindly provided by Prof. Spyros Pournaras, School of Medicine, National and Kapodistrian University of Athens). Bacteria were routinely grown in Mueller–Hinton Broth (Lab M, Greater Manchester, UK) or Mueller–Hinton agar (Lab M, Greater Manchester, UK) at 37 °C.

#### 3.5.2. Determination of Minimum Inhibitory Concentration (MIC)

Determination of MICs was performed in sterile 96 well polystyrene microtiter plates (Kisker, Steinfurt, Germany) according to Tsavea and Mossialos [95], with some minor modifications. Briefly, overnight bacterial cultures cultured in Mueller–Hinton broth were adjusted to 0.5 McFarland turbidity standard (approx. 1.5 × 10^8^ CFU mL^−1^). Approximately 5 × 10^4^ CFUs in 10 μL Mueller–Hinton broth were added to 190 μL Mueller–Hinton broth containing the tested compound at 30, 25, 20, 15, 10 and 5 mg/mL in triplicates. Control wells contained Mueller–Hinton broth inoculated with bacteria. Optical density (OD) was determined at 630 nm using an ELx808 Absorbance Microplate Reader (BioTek, Santa Clara, CA, USA) just prior to incubation (t = 0) and after 24 h incubation (t = 24) at 37 °C. The OD for each replicate well at t = 0 was subtracted from the OD of the same replicate well at t = 24. The growth inhibition at each compound concentration was determined using the following formula: % inhibition = 1 − (OD test well/OD of corresponding control well) × 100. Minimum inhibitory concentration was determined as the lowest concentration which resulted in 100% growth inhibition.

#### 3.5.3. Determination of Minimum Bactericidal Concentration (MBC)

Minimum bactericidal concentration (MBC) is the lowest concentration of any compound killing tested bacteria. The MBC was determined by transferring a small quantity of sample contained in each replicate well of the microtiter plates to Mueller–Hinton agar plates by using a microplate replicator (Boekel Scientific, Feasterville, PA, USA). The plates were incubated at 37 °C for 24 h. The MBC was determined as the lowest concentration at which no grown colonies could be observed [96].

### 3.6. INFOGEST Protocol

In order to investigate the effect of the digestive tract on the oxidative stability of the beverage, the INFOGEST protocol was used according to Floros et al. [97]. According to the INFOGEST protocol, digestion involves the exposure of food to three phases (oral, gastric, and intestinal phase). The electrolytes used for every stage were prepared in advance in stock solutions and stored at −10 °C. Specifically, stock solutions of KCl (0.5 M), KH_2_PO_4_ (0.5 M), NaHCO_3_ (1 M), NaCl (2 M), MgCl_2_(H_2_O)_6_ (0.15 M), (NH_4_)_2_CO_3_ (0.5 M), HCl (0.09 M), and CaCl_2_(H_2_O)_2_ (0.025 M) were prepared. These stock solutions were used to create the simulated fluids (1.25×) for each stage of digestion, known as simulated salivary fluid (SSF), simulated gastric fluid (SGF), and simulated intestinal phase (SIF), as described sufficiently in the INFOGEST protocol manuscript [98].

#### 3.6.1. Oral Digestion Phase

The protocol began with the preparation of the samples and their primary homogenization. One gram of sample was added to a test tube and mixed with SSF (1.25×). Distilled water was added to achieve a final volume ratio of 1:1. The final mixture was transferred to a heated incubator, where the test tube was shaken under heating for 2 min at a constant temperature of 37 °C.

#### 3.6.2. Gastric Digestion Phase

The oral bolus was mixed with SGF (1.25×). Additionally, pepsin was solubilized with water to reach a final activity of 2000 U/mL and added to the mixture. The pH was set to 3 by the addition of the HCl solution (1 M). Distilled water was added until a final volume ratio of 1:1 was reached. The final mixture was transferred to the temperature-controlled incubator, where it remained for 2 h at a temperature of 37 °C.

#### 3.6.3. Intestinal Digestion Phase

The gastric chyme was mixed with SIF (1.25×), pancreatin solution (100 TAME U/mL), and bile salt solution (10 mmol/L). The pH was set to 7 by the addition of NaOH solution (1 M). Distilled water was added until a final volume ratio of 1:1 was reached. The mixture was finally transferred to the temperature-controlled incubator, where it remained for 2 h at a constant temperature of 37 °C.

#### 3.6.4. Sample Treatment and Storage

BHT (500 ppm) was added after the end of the protocol to inhibit further oxidation, while samples were frozen and kept at −20 °C until further evaluation.

#### 3.6.5. DPPH Assay (% DPPH Inhibition)

Antioxidant activity was measured as % DPPH inhibition as described by Tepe et al. [99] with slight modifications. An amount of 0.00192 % *w*/*v* DPPH solution was prepared by diluting 0.48 mg DPPH (2,2-Diphenyl-1-picrylhydrazyl) with 25 mL methanol. An amount of 0.1 mL of the beverage was added to 3.9 mL of the methanol DPPH solution. The samples were incubated for 30 min in the dark, and the absorbance was read against a blank at 517 nm with spectrophotometer (Unicam, Helios γ). The % DPPH inhibition was calculated by the following equation:$$Scavenging\;capacity\%=\frac{A_{DPPH}-A_{sample}}{A_{DPPH}}\times 100\;$$
where *A_DPPH_* is the absorbance of the control reaction (containing all reagents except the sample), and *A_sample_* is the absorbance of the sample.

#### 3.6.6. Ferric Reducing Antioxidant Power (FRAP)

The reducing power of the powder was determined according to the method of Oyaizu [100] with slight modifications. Briefly, the beverage was diluted with distilled water (final ratio: 1/40), and 1 mL of this solution was mixed with 2 mL distilled water and 2.5 mL of 0.2M phosphate buffer (pH 6.6). Then, 2.5 mL of 1% potassium ferricyanide was added, and the sample was incubated at 50 °C for 30 min. Afterwards, 2.5 mL of 10% trichloroacetic acid was added to the mixture, and it was centrifuged at 2000 rpm for 10 min. A total of 2.5 mL of the supernatant were collected and mixed with 2.5 mL distilled water and 0.5 mL of 0.1% ferric chloride. The absorbance was read at 700 nm with a spectrophotometer (Unicam, Helios γ).

### 3.7. Sensory Acceptance Testing

The sensory acceptance testing was conducted according to the method by O’Sullivan [101] with modifications. The test was performed at the food laboratory and 31 volunteers (16 males, 15 females) who regularly engage in physical exercise (at least 1–2/week) and consume sports or energy drinks were recruited from the University of Thessaly. The sensory analysis was done to evaluate the sensory acceptability of the beverage. The test was conducted using a 9-point hedonic scale, ranging from 1 for “dislike extremely” to score 9 “like extremely”. Each assessor was provided with approximately 50 mL of the beverage and 250 mL of water for mouth rinsing before sample testing. The sensory attributes evaluated were taste, aftertaste, color, aroma, solubility, mouthfeel, acidity, sweetness, and overall acceptance. In addition, the assessors filled out a short questionnaire regarding their demographic characteristics, the frequency at which they consume sports/energy drinks, and the frequency at which they engage in physical activity/exercise. After the testing, they were also asked to describe the flavor of the beverage and their willingness to purchase the beverage.

### 3.8. Statistical Analysis

All experiments were performed in triplicate, and all data are expressed as the mean ± standard deviation. Differences were considered significant at *p* < 0.05. All statistical analyses were performed with the SPSS software (version 14.0; SPSS). For statistical analysis, one-way ANOVA was applied followed by Dunnett’s test for multiple pair-wise comparisons. Dose response relationships were examined by Spearman’s correlation analysis.

## 4. Conclusions

This study aimed to evaluate the antioxidant and antibacterial properties of a functional sports beverage formulation. The total phenolic content assessed by the Folin–Ciocalteu assay was found to be 7.58 ± 0.066 mg GAE/mL, while the main phenolic compounds identified by HPLC were catechin, epicatechin, protocatechuic acid, luteolin 7-glucoside, and kaempferol-3-O-β-rutinoside. Furthermore, the antioxidant analyses showed that the beverage possessed strong antioxidant properties. TPC was found to be strongly correlated with TAC (R^2^ = 896), while the simulated gastrointestinal conditions showed that digestion positively affected the oxidative stability of the beverage and increases its antioxidant properties. In addition, the beverage exhibited bacteriostatic effects against *Staphylococcus aureus*, a Gram-positive pathogen, and *Pseudomonas aeruginosa*, a Gram-negative opportunistic pathogen. Lastly, the sensory acceptance test revealed that the functional sports beverage was highly accepted by the assessors. A total of 96.8% of the assessors reported being willing to buy the beverage and that they would prefer it over others on the market.

The present study is not without limitations. Specifically, in this study, we did not directly assess the potential ergogenic or anti-fatigue properties of the formulated beverage. Additionally, the antioxidant properties of the beverage were only assessed in vitro. Further experiments and clinical trials should be conducted in order to evaluate the ergogenic effects and confirm the antioxidant properties of this beverage in athletes.

## Figures and Tables

**Figure 1 ijms-24-03558-f001:**
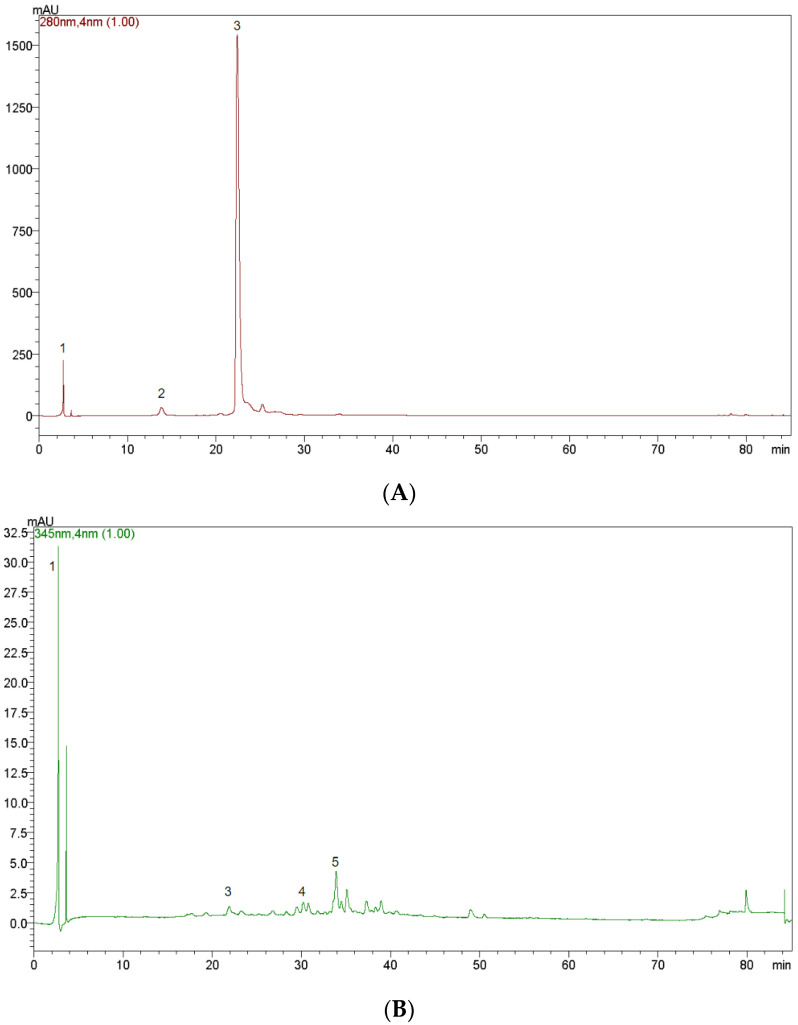
HPLC chromatograms of the sports beverage at 280 nm (**A**) and 345 nm (**B**). The peaks represent the following phenolics: 1 = epicatechin, 2 = protocatechuic acid, 3 = catechin, 4 = luteolin 7-glucoside, and 5 = kaempferol-3-O-β-rutinoside. Note that not all phenolic compounds showed peaks at both wavelengths tested.

**Figure 2 ijms-24-03558-f002:**
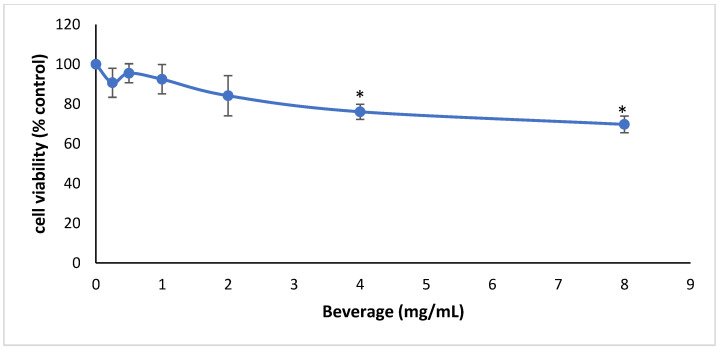
Cell viability following the treatment of MSC cells with the sports beverage. The results are presented as the means ± SEM of three independent experiments carried out in triplicate. * *p* < 0.05 indicates significant difference from the control value.

**Figure 3 ijms-24-03558-f003:**
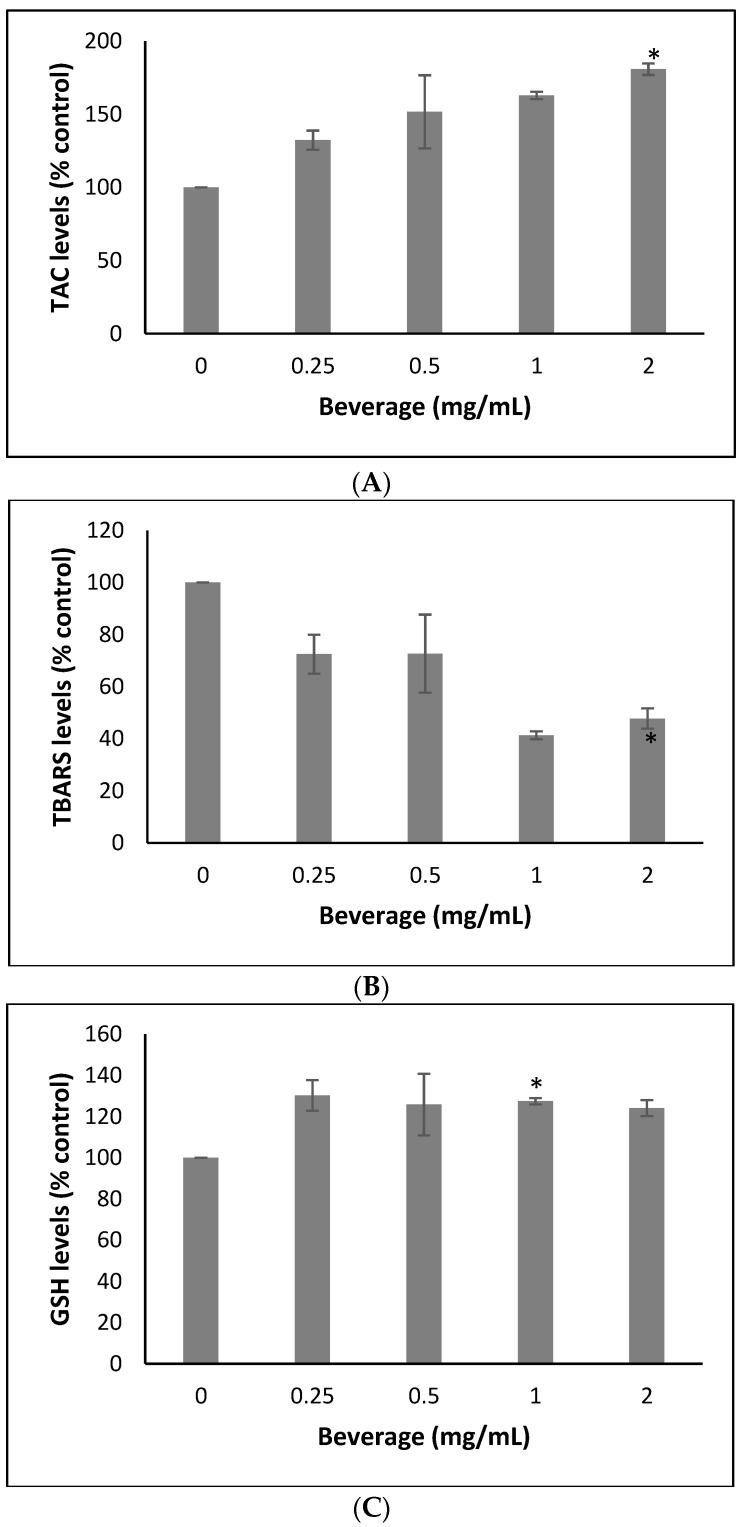
Effects of the sports beverage on (**A**) TAC, (**B**) TBARS levels and (**C**) GSH levels. All values are presented as the mean ± SD of three experiments. * means that levels were statistically significant compared to control (*p* < 0.05).

**Figure 4 ijms-24-03558-f004:**
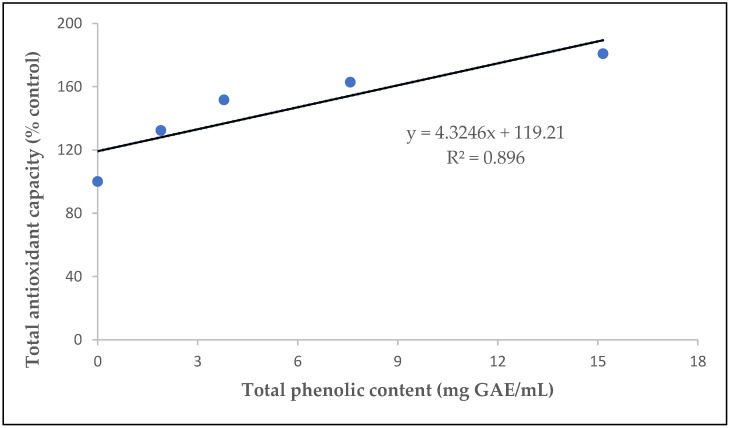
Correlation between the total phenolic content and total antioxidant capacity of the sports beverage.

**Figure 5 ijms-24-03558-f005:**
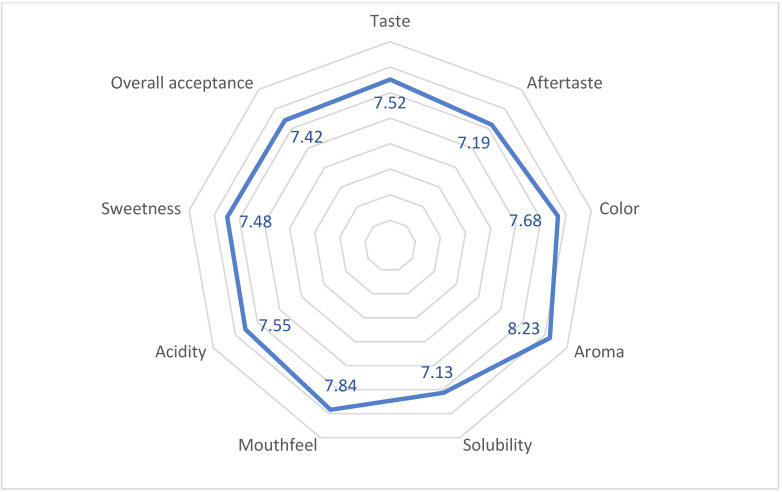
Sensory profile of the sports beverage.

**Table 1 ijms-24-03558-t001:** Polyphenolic composition of the sports beverage.

Polyphenolic Compound	Concentration (mg/mL)
Catechin	2.149
Epicatechin	0.024
Protocatechuic acid	0.012
Luteolin 7-glucoside	0.001
Kaempferol-3-O-β-rutinoside	0.001

Table values are means of triplicate determinations.

**Table 2 ijms-24-03558-t002:** The antibacterial activity of the sports beverage.

Bacterial Species	MIC (mg/mL)	MBC (mg/mL)
*Staphylococcus aureus*	30	>30
*Pseudomonas aeruginosa*	30	>30

MIC: minimum inhibitory concentration; MBC: minimum bactericidal concentration.

**Table 3 ijms-24-03558-t003:** The oxidative stability (% DPPH inhibition, FRAP) of the sports beverage.

	% DPPH Scavenging	FRAP (μΜ TE/mL)
Oral digestion phase (t = 0)	70.8 ± 1.7 ^a^	728 ± 93 ^a^
Gastric digestion phase (t = 135 min)	74.8 ± 1.3 ^b^	804 ± 180 ^a^
Intestinal digestion phase (t = 255 min)	88.1 ± 0.3 ^c^	1855 ± 72 ^b^

Table values are means ± standard deviations (*n* = 3). Different superscript letters in the same column represent statistical differences (*p* ≤ 0.05).

**Table 4 ijms-24-03558-t004:** Questionnaire results.

Question	N	%
**Age Group**		
18–29	21	67.7
30–44	7	22.6
45+	3	9.7
**Do you engage in exercise or physical activity?**		
Yes	31	100
No	0	0
**How often do you engage in exercise or physical activity?**		
1–2/week	17	54.8
3–5/week	11	35.5
5+/week	3	9.7
**Do you consume energy drinks or sports drinks?**		
Yes	30	96.8
No	1	3.2
**If yes, how often do you consume energy drinks or sports drinks?**		
Rarely (a few times per year)	9	29.0
Sometimes (a few times per month)	13	41.9
Often (1–3/week)	7	22.6
Daily	1	3.2

## Data Availability

The data presented in this study are available upon request from the corresponding author.

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
