# Peer review of "Antioxidant and Antibacterial Properties of a Functional Sports Beverage Formulation"

_ijms, 2023, doi:10.3390/ijms24043558_

Round 1

Reviewer 1 Report

General comments

Table format: Please organize the table format all through the manuscript.  

Discussion: when discussing the antioxidant capacity in the cell, it is recommended to consider the bioavailability or bioaccessibility of target components.

Specific comments:

Part 2.2: How many replicas were run for each sample, and how were the statistics doing? For the result in Table 2, my suggestion is to just keep the concentration in beverages (mg/mL). It caused confusion to see the data in another column.

Part 2.5: What level were the total phenolic compounds and individual phenolic compounds in the different digestion stages?

Part 2.6: The same results are shown in Figure 5 andTable 5. Keeping only one is fine.

Line 489: Was the cell treated with powder or beverage? Same question for Part 3.6? What’s the sample used? Please specify it in the method section.

Author Response

General comments

Table format: Please organize the table format all through the manuscript. 

The table format has been organized throughout the manuscript.

Discussion: when discussing the antioxidant capacity in the cell, it is recommended to consider the bioavailability or bioaccessibility of target components.

Thank you for your comment. We agree that when a food supplement is examined the bioavailability of its compounds constitute an important issue. So, as the reviewer has suggested, we added information regarding the bioavailability of the beverage’s compounds. Specifically, the following paragraph was written:

“When examining bioactivities of food or beverages, the absorption and the bioavailability of their bioactive compounds are an important issue. Several bioactive compounds have been found to be effective in in vitro studies, but at much higher concentrations than those of their peak levels in human plasma [Narumi et al., 2014; Cai et al., 2018]. In the present study, the maximum concentration of the beverage in the form of powder used in the cell culture experiments was 2 mg/mL. The main polyphenolic compounds identified in the beverage were catechin, epicatechin, protocate-chuic acid, luteolin-7-glucoside and kaempferol-3-O-rutinoside. According to the amount of these polyphenols in the beverage, their concentrations in the cell culture medium at 2 mg/mL of the beverage in the form of powder were: catechin (150 μΜ), epicatechin (1.8 μΜ), protocatechuic acid (1.7 μΜ), luteolin-7-glucoside (0.047 μΜ) and kaempferol-3-O-rutinoside (0.034 μΜ). The concentration of epicatechin was physiologically meaningful, since it has been shown to achieve 0.6 μΜ in human plasma after administration of green tea [Yang et al., 1998]. Protocatechuic acid’s con-centration can also be achieved in the human organism, since this phenolic acid has been reported to remain stable in human plasma at 30 μΜ for 24 h [Song et al., 2020]. Moreover, the concentration of luteolin-7-glucoside was also meaningful, since a bioavailability study has reported that it absorbed quickly and reached levels of about 0.7 μΜ in rat plasma [Yasuda et al., 2015]. Regarding kaempferol-3-O-rutinoside’s bioavailability, studies have shown that kaempferol glucosides are mainly absorbed through mechanisms of active transport and deglycosylation [Kashyap et al., 2017]. Although, there has not been much data of bioavailability of kaempferol and its glycosides forms, a concentration of about 0.05 mM of this polyphenol has been found in human plasma [Kashyap et al., 2017]. Thus, our concentration of kaempferol-3-O-rutinoside of 0.034 μΜ may be found in the human organism. Catechin’s concentration of 150 μΜ in our beverage formulation was higher than that reported in human plasma after consumption of beverages being rich in this polyphenol [Narumi et al., 2014]”.

Specific comments:

Part 2.2: How many replicas were run for each sample, and how were the statistics doing? For the result in Table 2, my suggestion is to just keep the concentration in beverages (mg/mL). It caused confusion to see the data in another column.

All experiments were performed in triplicate and all data are expressed as the mean ± standard deviation. Differences were considered significant at p<0.05. All statistical analyses were performed with the SPSS software (version 14.0; SPSS).  

The concentrations in the powder form have been deleted and the concentrations in the beverage (mg/mL) have been kept, according to your suggestion.

Part 2.5: What level were the total phenolic compounds and individual phenolic compounds in the different digestion stages?

We did not measure total phenolics, but there are several references in the literature (included in the manuscript) that can account for the antioxidative behavior of the beverage. Besides, the aim was to study the oxidative stability of the beverage and not the fate of individual phenolic compounds during digestion. This is a good idea that can be explored in the next phase of the experiments.

Part 2.6: The same results are shown in Figure 5 and Table 5. Keeping only one is fine.

Table 5 has been deleted. Thank you for your comment.

Line 489: Was the cell treated with powder or beverage? Same question for Part 3.6? What’s the sample used? Please specify it in the method section.

This has been corrected. The sample used was the beverage formulation in the form of powder diluted in water for Part 3.4.3. For Part 3.6 the beverage was used.

Reviewer 2 Report

This work aimed to develop a formulation of a functional sport beverage with ergogenic and antioxidant properties and evaluate its physiochemical properties including oxidation stability, antioxidant, antibacterial activity, and sensory acceptability. The results would provide valuable base for the production of novel functional sport beverage. However, some revisions need to be done before being accepted by International Journal of Molecular Sciences.

-L23, the results of TBARS, TAC and GSH are not depicted in the abstract section.

-L25, please give out the full name of INFOGEST.

-L164, the table is so simple and should be deleted.

-L272, delete “in”.

-L284, what is the accurate MBC?

-L335, 337, Figure 5 and Table 5 are repeated, and one of both should be remained.

-L370-372, please rewrite this sentence.

-L382-385, too long sentence.

-L386-390, too long sentence.

-L669, the reference format should be revised carefully based on the requirement of journal. Why to cite so many references in a research paper?

-Some grammar errors should be checked and revised in the whole manuscript.

Author Response

This work aimed to develop a formulation of a functional sport beverage with ergogenic and antioxidant properties and evaluate its physiochemical properties including oxidation stability, antioxidant, antibacterial activity, and sensory acceptability. The results would provide valuable base for the production of novel functional sport beverage. However, some revisions need to be done before being accepted by International Journal of Molecular Sciences.

-L23, the results of TBARS, TAC and GSH are not depicted in the abstract section.

The results of TBARS, TAC and GSH have been added to the abstract section and the abstract has been rewritten to:

“Athletes often consume functional beverages in order to improve performance and reduce oxidative stress, caused by high-intensity exercise. The present study aimed to evaluate the antioxidant and antibacterial properties of a functional sports beverage formulation. The beverage’s antioxidant effects were assessed on human mesenchymal stem cells (MSCs) by determining thiobarbituric acid reactive substances (TBARS; TBARS levels decreased significantly by 52.67% at 2.0 mg/mL), total antioxidant capacity (TAC; TAC levels increased significantly by 80.82% at 2.0 mg/mL) and reduced glutathione (GSH; GSH levels increased significantly by 24.13% at 2.0 mg/mL) levels. Furthermore, the beverage underwent simulated digestion following the INFOGEST protocol to assess its oxidative stability. The analysis of the total phenolic content (TPC) using the Folin-Ciocalteu assay revealed that the beverage contains a TPC of 7.58 ± 0.066 mg GAE/mL, while the phenolics identified by HPLC were catechin (2.149 mg/mL), epicatechin (0.024 mg/mL), protocatechuic acid (0.012 mg/mL), luteolin 7-glucoside (0.001 mg/mL), and kaempferol‑3‑O‑β‑rutinoside (0.001 mg/mL). The beverage’s TPC was strongly correlated with TAC (R2 = 0,896). Moreover, the beverage showcased inhibitory and bacteriostatic effects against Staphylococcus aureus and Pseudomonas aeruginosa. Lastly, the sensory acceptance test demonstrated that the functional sports beverage was well accepted by the assessors”.

-L25, please give out the full name of INFOGEST.

To our knowledge INFOGEST is the full name that is used in the literature (INFOGEST is an international network of excellence on the fate of food in the gastrointestinal tract (https://www.cost-infogest.eu/)).

-L164, the table is so simple and should be deleted.

This table has been deleted.

-L272, delete “in”.

“in” has been deleted.

-L284, what is the accurate MBC?

MIC and MBC were determined up to 30 mg/ml due to solubility limitations of the tested compound. Above 30 mg/ml solubility was poor and the results would not be reliable. However, it is clear the bacteriostatic effect against both tested bacteria at 30 mg/ml (MIC). At this concentration no bactericidal effect was observed therefore we assume that MBC is higher than 30 mg/ml.

-L335, 337, Figure 5 and Table 5 are repeated, and one of both should be remained.

Table 5 has been deleted. Thank you for your comment.

-L370-372, please rewrite this sentence.

This sentence has been rewritten to: “In addition, the formulated beverage contains tripotassium citrate, trisodium citrate and magnesium citrate in order to achieve optimal pre-hydration, according to the recommendations of the ACSM”.

-L382-385, too long sentence.

This sentence has been rewritten to: “In addition, cocoa extract was added to the beverage, which has been shown to be able to reduce exercise-induced oxidative stress, due to the improvement of the nitric oxide function”.

-L386-390, too long sentence.

This sentence has been rewritten to: “Lastly, the beverage contains rosehip powder and mixed berries powder. Fruit-derived polyphenols have been shown to improve exercise performance and enhance recovery by improving vascular function via NO-mediated mechanisms, and by decreasing oxidative stress via signaling through the nuclear factor erythroid 2–related factor 2 (Nrf2)”.

-L669, the reference format should be revised carefully based on the requirement of journal. Why to cite so many references in a research paper?

The reference format has been revised according to the requirements of the journal. Due to the beverage consisting of several compounds, we think that it is necessary to include all these references in order to provide all the necessary information and evidence.

-Some grammar errors should be checked and revised in the whole manuscript.

Grammar errors have been corrected throughout the manuscript.

Reviewer 3 Report

The research work entitled "Antioxidant and antibacterial properties of a functional sports beverage formulation" is a hypothesis driven work and the article is well-written. I have some suggestions regarding the methodology portions and therefore I recommend some changes in the manuscript. The authors are also advised to make some modifitions in the writing part of the manuscript. Please find my comments below;

1. The abstract may contain more information on the methodology and quantitative data of the study. Otherwise, the abstract is well-structured.

2. Introduction section may contain information on the antibiotic resistance concept, as they analyzed the antimicrobial potentials in resistant strians.

3. Authors analyzed the total polyphenol content; I suggest to include the results of total flavonoid content (can be estimated using aluminium chloride method)

4. HPLC analysis of the phenolic components are fine; however, I recommed to include the LCMS analysis in the work for a better clarity in the composition aspects. (incase HPLC alone is shown, authors need to include limit of detection and limit of quantification in the data)

5. The figure 2 (cell viability) may be represented as a line graph. It can better communicate the toxicity result.

6. The MIC values must be repeated and represented with standard deviation. Also, I suggest to include the disc diffusion assay that can provide the zone of inhibition.

7. The questionnaire seems to be too preliminary; only 31 recordings are not sufficient to explain the results.

8. Authors need to mention the inclusion and exclusion criteria for the selection of 31 representatives. Because, age group and even disease conditions can influence the sensory acceptance of the product.

Author Response

The research work entitled "Antioxidant and antibacterial properties of a functional sports beverage formulation" is a hypothesis driven work and the article is well-written. I have some suggestions regarding the methodology portions and therefore I recommend some changes in the manuscript. The authors are also advised to make some modifitions in the writing part of the manuscript. Please find my comments below;

  1. The abstract may contain more information on the methodology and quantitative data of the study. Otherwise, the abstract is well-structured.

Thank you for your comment. In order to contain more information on the methodology and quantitative data, the abstract has been rewritten to:

“Athletes often consume functional beverages in order to improve performance and reduce oxidative stress, caused by high-intensity exercise. The present study aimed to evaluate the antioxidant and antibacterial properties of a functional sports beverage formulation. The beverage’s antioxidant effects were assessed on human mesenchymal stem cells (MSCs) by determining thiobarbituric acid reactive substances (TBARS; TBARS levels decreased significantly by 52.67% at 2.0 mg/mL), total antioxidant capacity (TAC; TAC levels increased significantly by 80.82% at 2.0 mg/mL) and reduced glutathione (GSH; GSH levels increased significantly by 24.13% at 2.0 mg/mL) levels. Furthermore, the beverage underwent simulated digestion following the INFOGEST protocol to assess its oxidative stability. The analysis of the total phenolic content (TPC) using the Folin-Ciocalteu assay revealed that the beverage contains a TPC of 7.58 ± 0.066 mg GAE/mL, while the phenolics identified by HPLC were catechin (2.149 mg/mL), epicatechin (0.024 mg/mL), protocatechuic acid (0.012 mg/mL), luteolin 7-glucoside (0.001 mg/mL), and kaempferol‑3‑O‑β‑rutinoside (0.001 mg/mL). The beverage’s TPC was strongly correlated with TAC (R2 = 0,896). Moreover, the beverage showcased inhibitory and bacteriostatic effects against Staphylococcus aureus and Pseudomonas aeruginosa. Lastly, the sensory acceptance test demonstrated that the functional sports beverage was well accepted by the assessors”.

  1. Introduction section may contain information on the antibiotic resistance concept, as they analyzed the antimicrobial potentials in resistant strians.

The following paragraph has been now included in the Introduction: “The use of antibiotics – including the over- and misuse – in human and veterinary practices inevitably selected for resistant pathogens the last decades. Therefore, antibiotic resistance is now considered as one of the top health challenges facing the 21st century and called the World Health Organization (WHO) into action (Meyer et al. 2013). Currently, diverse natural products (including functional foods like honey) demonstrating antibacterial activity have been widely investigated as alternatives in order to prevent infections or to combat multidrug resistant pathogens (Kafantaris et al. 2021)”.

  1. Authors analyzed the total polyphenol content; I suggest to include the results of total flavonoid content (can be estimated using aluminium chloride method)

          Thank you for your comment. The authors undertake to conduct additional experiments in order to         determine the concentration of flavonoids in their future research.

  1. HPLC analysis of the phenolic components are fine; however, I recommed to include the LCMS analysis in the work for a better clarity in the composition aspects. (incase HPLC alone is shown, authors need to include limit of detection and limit of quantification in the data)

Τhe phenolics identified by HPLC were catechin (23.01 mg/g), epicatechin (0.26 mg/g), protocatechuic acid (0.13 mg/g), luteolin 7-glucoside (0.01 mg/g), and kaempferol‑3‑O‑β‑rutinoside (0.01 mg/g). Limit of detection (LOD) and limit of quantification (LOQ) were established experimentally, using the ratio signal-to-noise methodology. Due to the fact that all considered compounds except kaempferol‑3‑O‑β‑rutinoside appeared in chromatogram with high peaks the LOD and LOQ were determined only for kaempferol‑3‑O‑β‑rutinoside (0.01 mg/g) and the results obtained for these limits were 0.002 mg/g and 0.006 mg/g, respectively.

  1. The figure 2 (cell viability) may be represented as a line graph. It can better communicate the toxicity result.

As the reviewer has suggested, the cell viability has been presented as a line graph. We agree that this graph is better than the previous one. Thank you for your comment.

  1. The MIC values must be repeated and represented with standard deviation. Also, I suggest to include the disc diffusion assay that can provide the zone of inhibition.

By definition Minimum Inhibitory Concentration is the lowest concentration of any tested compound which results in 100% growth inhibition. In this study MIC determination has been performed in triplicates for each tested concentration. MIC was determined as the concentration that no growth was observed in all three replicas (100 % inhibition). Therefore, SD is not meaningful. On the other hand, SD should be included if disc diffusion assay was performed (zones of inhibition in most cases are indeed variable). However, determination of MIC in broth (the method used in this study) is generally regarded as a more sensitive and quantitatively precise method to study antimicrobial activity compared to disc diffusion assay because diffusion rates of active substances might be slower in agar than in broth.

  1. The questionnaire seems to be too preliminary; only 31 recordings are not sufficient to explain the results.

The methodology used for the sensory acceptance testing was according to O'Sullivan (Chapter 3 - Sensory Affective (Hedonic) Testing, in A Handbook for Sensory and Consumer-Driven New Product Development), based on which “sensory acceptance testing can be employed during the development and optimization processes as a means of assessing variant suitability in a hedonic fashion and involves anywhere from 25 to 75 individuals”. In future experiments, consumer testing will be conducted with larger numbers of consumers.

  1. Authors need to mention the inclusion and exclusion criteria for the selection of 31 representatives. Because, age group and even disease conditions can influence the sensory acceptance of the product.

In order to better explain the criteria for selecting the assessors for the sensory acceptance test, the following has been added in Part 3.7.: “The test was performed at the food laboratory and 31 volunteers (16 males, 15 females) who regularly engage in physical exercise (at least 1-2/week) and consume sports or energy drinks were recruited from the University of Thessaly”. In addition, the age groups have now been added in Table 4.

Reviewer 4 Report

The authors prepared a functional sports beverage formulation, without added sugars, and analyzed its antioxidant and antibacterial properties.

Beverage antioxidant effects were assessed on human mesenchymal stem cells (MSCs) by determining thiobarbituric acid reactive substance (TBARS), total antioxidant capacity (TAC) and reduced glutathione (GSH) levels. In addition, the beverage underwent simulated digestion following the INFOGEST protocol to assess its oxidative stability.

The total phenolic content (TPC) was analyzed and catechin, epicatechin, protocatechuic acid, luteolin 7-glucoside, and kaempferol-3-O-β-rutinoside were identified by HPLC.

The TPC of the beverage was found to be strongly correlated with total antioxidant capacity (TAC). Inhibitory and bacteriostatic effects against Staphylococcus aureus and Pseudomonas aeruginosa were also investigated.

Additionally, a sensory acceptance test was favorable with a mean score for the overall acceptance of approximately 7.4.

In my opinion this research is well performed and the article is well written and can be published with minor corrections in the special issue "The Effect of Phenolic Compounds in Human Diseases".

I have only one observation to make: references 18 and 20, 19 and 21, 12 and 22 are identical.

Author Response

The authors prepared a functional sports beverage formulation, without added sugars, and analyzed its antioxidant and antibacterial properties.

Beverage antioxidant effects were assessed on human mesenchymal stem cells (MSCs) by determining thiobarbituric acid reactive substance (TBARS), total antioxidant capacity (TAC) and reduced glutathione (GSH) levels. In addition, the beverage underwent simulated digestion following the INFOGEST protocol to assess its oxidative stability.

The total phenolic content (TPC) was analyzed and catechin, epicatechin, protocatechuic acid, luteolin 7-glucoside, and kaempferol-3-O-β-rutinoside were identified by HPLC.

The TPC of the beverage was found to be strongly correlated with total antioxidant capacity (TAC). Inhibitory and bacteriostatic effects against Staphylococcus aureus and Pseudomonas aeruginosa were also investigated.

Additionally, a sensory acceptance test was favorable with a mean score for the overall acceptance of approximately 7.4.

In my opinion this research is well performed and the article is well written and can be published with minor corrections in the special issue "The Effect of Phenolic Compounds in Human Diseases".

I have only one observation to make: references 18 and 20, 19 and 21, 12 and 22 are identical.

Thank you for your comments. The identical references have been corrected.

Round 2

Reviewer 3 Report

No more comments